# Nature-Based Rehabilitation for Patients with Long-Standing Stress-Related Mental Disorders: A Qualitative Evidence Synthesis of Patients’ Experiences

**DOI:** 10.3390/ijerph18136897

**Published:** 2021-06-27

**Authors:** Anna Bergenheim, Gunnar Ahlborg, Susanne Bernhardsson

**Affiliations:** 1Region Västra Götaland, Research, Education, Development & Innovation, Primary Health Care, Sweden; susanne.bernhardsson@vgregion.se; 2Department of Health and Rehabilitation, Sahlgrenska Academy, Institute of Neuroscience and Physiology, University of Gothenburg, 413 46 Gothenburg, Sweden; 3Region Västra Götaland, Institute of Stress Medicine, 413 19 Gothenburg, Sweden; gunnar.ahlborg@vgregion.se; 4Region Västra Götaland, HTA-Centrum, Sahlgrenska University Hospital, 413 46 Gothenburg, Sweden

**Keywords:** stress-related disorders, nature-based rehabilitation, qualitative, synthesis

## Abstract

Stress-related mental disorders contribute to work disabilities globally and are a common cause for sick leave. Nature-based rehabilitation (NBR) is a multi-disciplinary approach offered to this patient group on a limited scale. Qualitative studies provide insight into patients’ experiences of NBR, and there is a need to synthesize and assess the certainty of evidence for patient-experienced benefits. The aim was to identify, appraise, and synthesize studies reporting experiences and perceived benefits of participation in multidisciplinary, group-based NBR of adult patients with long-standing stress-related mental disorders. PubMed, Embase, CINAHL, AMED, APA PsycInfo, and the Cochrane Library were searched from inception to December 2020. Reference lists of relevant publications were searched. After title and abstract screening, full-text articles were retrieved and assessed for inclusion. The methodological quality of the included studies was assessed, and certainty of evidence was appraised according to CERQual. The search yielded 362 unique records; 19 full-text publications were assessed for eligibility, and 5 studies were included in the synthesis. The studies were considered relevant regarding context, population, and intervention, and quality was generally assessed as moderate to high. Extracted texts were inductively coded and organized into 16 descriptive themes and 4 broad, analytical themes: Instilling calm and joy; Needs being met; Gaining new insights; and Personal growth. Experiences and perceived benefits of participating in NBR and spending time in a nature environment were described as positive for recovery. Nine of the descriptive themes were based on explicit results from at least four of the five studies. Confidence in the evidence of the qualitative findings ranged from moderate to low. Moderate-to-low certainty evidence from the included studies suggests that patients with long-standing stress-related mental disorders experience positive health effects from participating in NBR.

## 1. Introduction

Mental health problems are estimated to be one of the major contributors to work disabilities globally [1,2]. Long-standing stress-related mental disorders comprise several conditions, and there is a variation between countries in clinical assessment, diagnoses used, and treatment/rehabilitation practices. The most common diagnoses are anxiety, depression, and exhaustion/burnout, depending not only on the clinical picture but also on the national or local clinical practice [3]. In Sweden, the prevalence of sick leave caused by mental health problems increased around the turn of the millennium [4,5], and stress-related mental illness is today the most common cause of sick leave from work [6], making it important also from a societal perspective to find effective treatment/rehabilitation.

In Sweden, most patients with stress-related mental disorders receive individualized treatment in primary care or occupational healthcare service. The provided care varies depending on local practices and resources, but may include a combination of psychological support, physical activity, behavioral therapy, relaxation exercise, or group-based patient education (e.g., sleep and stress), if needed in combination with medication for symptoms of depression and anxiety. Patients with severe conditions often receive specialized psychological treatments, occupational therapy, physiotherapy, and psycho-education. This type of treatment and rehabilitation is typically provided by multidisciplinary teams.

One such multidisciplinary approach is nature-based rehabilitation (NBR), a specific form of a stress management course in which nature and garden play an important role, and rehabilitation to a large extent takes place in an outdoor setting [7]. However, NBR is neither internationally nor nationally standardized nor has a clear-cut definition, but is based on the presumed stress-reducing effects inherent with nature and garden environments, and the use of activities that enhance such effects and promote well-being in nature [8,9]. The NBR programs can be similar to other multidisciplinary program s in content, with components such as psycho-educative talks about how to prevent stress, bodily exercises, mindfulness, and creative workshops. In Sweden, NBR typically involves a multi-disciplinary rehabilitation team, usually including a physiotherapist, an occupational therapist, a psychotherapist/psychologist, and personnel with competences related to the garden and nature. Each profession contributes to the content of the programs on an interdisciplinary basis. The duration of NBR programs varies from a few months to up to a year and includes both individual and group-based activities. The intensity of NBR usually ranges between two and four hours per day, two to four days per week. The NBR activities include mild and limited sensory stimulation and therapeutic activities in a specially designed garden or selected nature environment.

Research is limited on the effectiveness of NBR interventions for patients with stress-related disorders. A recent systematic review of outdoor nature-based interventions for stress recovery suggests positive psychological and emotional effects after nature-based exposure [10]. However, only three of the included studies concerned patients with stress, and of those only one [7] evaluated a NBR intervention. That study showed positive effects on burnout, depression, anxiety, and well-being after participating in NBR. A recent health technology assessment (HTA) that focused on NBR for stress-related disorders identified ten studies, four of which were randomized controlled trials [11]. Based on those studies, the HTA report concluded that there was low certainty of evidence that NBR is no more effective than non-nature-based rehabilitation interventions, but that patients seemed to experience positive health effects.

Hence, evidence of the effect measured objectively in quantitative studies is limited, while patients subjectively report experiencing benefits from participating in NBR. There is today wide recognition of the need to draw evidence from different types of studies to inform decisions and policy making, and that qualitative studies are useful complements to effectiveness studies [12]. Qualitative studies on patients’ experiences of participating in NBR are therefore important to consider when assessing benefits of NBR interventions.

Synthesized qualitative findings of NBR experiences from the perspective of patients with stress-related mental disorders have to date not been published.

The aim of this systematic review of qualitative studies was to identify, appraise, and synthesize studies reporting experiences and perceived benefits of participating in multidisciplinary, group-based NBR, of adult patients with long-standing stress-related mental disorders.

## 2. Materials and Methods

### 2.1. Design

This is a systematic review and synthesis of qualitative studies, partially based on a HTA performed at HTA-centrum, Sahlgrenska University Hospital, in Gothenburg, Sweden [11]. This systematic review is reported according to the Enhancing transparency in reporting the synthesis of qualitative research statement (ENTREQ) [13].

### 2.2. Eligibility Criteria

Eligibility criteria were derived from the following Population, Exposure, Outcome model:Inclusion criteria:Population: Patients with long-standing (>6 months) stress-related mental disorders without known ongoing drug abuseExposure: Multidisciplinary, group-based NBROutcome: Experiences of participating in a NBR programStudy design: Qualitative studiesLanguages: English, Danish, Norwegian, Swedish

Exclusion criteria:

Post-traumatic stress disorder (PTSD), known drug abuse

### 2.3. Data Sources

Systematic literature searches were performed in PubMed, Embase, CINAHL, AMED, APA PsycInfo, and the Cochrane Library. Search terms and strategies were similar to those in the HTA, but modified to focus on patient experiences reported in qualitative studies. Searches were therefore performed from database inception to December 2020. A filter for qualitative studies, developed by the Canadian Health Libraries Association [14] and with some minor modifications/amendments for this search, was applied. Complete search strategies for all databases are presented in Appendix A. Reference lists of relevant articles were scrutinized for additional references.

### 2.4. Study Selection

There were three reviewers in total (A.B., G.A., and S.B.). Two reviewers (A.B. and S.B.) independently assessed abstracts and selected full-text articles for inclusion or exclusion. Disagreements were resolved in consensus. The remaining articles were assessed by all three reviewers, who read the articles independently of one another and decided in a consensus meeting which articles to include in the review.

### 2.5. Critical Appraisal and Confidence in Evidence

At least two reviewers critically appraised the included studies, using the 2014 version of the Swedish Agency for Health Technology Assessment and Assessment of Social Services (SBU) checklist for assessment of qualitative studies [15]. The SBU checklist comprises 21 questions covering the study’s aim, sampling, data collection, data analysis, and results presentation and transferability. Confidence in the evidence from the qualitative findings was assessed using the GRADE-CERQual approach [16]. The CERQual approach was developed by a GRADE working group and offers a similar, systematic, and transparent way of grading certainty of evidence in the findings from qualitative studies, as the GRADE approach does for quantitative studies. Confidence in each review finding is assessed in four domains (coherence, relevance, adequacy of data, and methodological limitations of the included studies) and is graded as high, moderate, low, or very low [16].

### 2.6. Data Extraction and Management

Data extraction from the included studies was performed by one reviewer (S.B.) and verified by another (A.B. or G.A.). Information about first author, study aim, setting, participants, age, gender, data collection method, and method of analysis was extracted. All text from the result sections in the included studies related to the objective of this review was extracted and entered verbatim into a work sheet.

### 2.7. Data Synthesis and Analysis

A qualitative synthesis of the findings from the included studies was performed. A thematic analysis was conducted of the extracted data using a qualitative synthesis methodology in three steps, as described by Thomas and Harden [17]. In the first step, the extracted text from the included studies was initially inductively coded line by line by one reviewer (S.B. or G.A.) and verified by another (G.A. or S.B.). The coding included identifying text segments that were relevant to the aims of the review, and deriving a code from these segments that captured the meaning and content of the segment. A coding scheme was created with all codes in consecutive order for each study. In the second step, these free codes were sorted and organized into related areas to construct descriptive themes. After the codes from the first study were mapped to a theme, any new concepts derived from subsequent studies were primarily coded into existing themes. New descriptive themes were created when necessary. During the analysis, comparisons of codes were constantly made within and across studies. In the third step, the findings were interpreted in relation to the review objective, and analytical themes were developed [17]. Throughout the process of analysis, disagreements were resolved in consensus among all reviewers, who discussed and revised the descriptive themes from which the final key review findings were formulated.

## 3. Results

### 3.1. Search Results

The database search and screening of reference lists identified 362 articles after removal of duplicates. After reading the abstracts, 358 articles were excluded. Nineteen articles were read in full text, and five studies were finally included that reported participants’ experiences of NBR [18,19,20,21,22]. A flowchart of the search results is presented in Figure 1.

### 3.2. Characteristics of Included Studies

Included studies, their aim, their setting, and their population are presented in Table 1. Excluded studies and reasons for exclusion are presented in Appendix A. The five included studies were published from 2012 to 2017, with the number of participants ranging from 5 to 43 (total number in all five studies was 94). The most recent study stemmed from a NBR project in the Nacadia Therapy Garden in Copenhagen, Denmark [22]. Three studies concerned the NBR program in the health garden at the Swedish University of Agricultural Sciences in Alnarp, Sweden [18,19,20]. One study reported experiences by participants in a NBR program at the Botanical Garden in Gothenburg (Gröna Rehab), Sweden [21]. All five studies used semi-structured interviews for data collection, and the method of analysis varied (Table 2).

### 3.3. Study Quality and Confidence in Evidence

The methods for data collection and analysis of the included studies are presented in Table 2, as well as their methodological quality, which was assessed as moderate to high. There were some limitations, however, mainly regarding the description of participants and researcher preunderstanding. Confidence in evidence for each of the review findings is presented in Table 3. Confidence varied from moderate to low.

### 3.4. Synthesis

The results of the qualitative synthesis and the CERQual assessment are presented in a summary of qualitative findings table (Table 3). Line-by-line coding of the included studies resulted in 178 preliminary codes, which reflected the meaning and content of the underlying text. These codes were organized into 16 descriptive themes and 4 broad, analytical themes (Table 4). The analytical themes went beyond the content of the original studies and related to the type of experiences and perceived effects of the nature and garden components of NBR: (1) Instilling calm and joy; (2) Needs being met; (3) Gaining new insights; (4) Personal growth. Sixteen key review findings were formulated based on the descriptive themes (Table 3).

### 3.5. Analytical Themes

#### 3.5.1. Instilling Calm and Joy

The first analytical theme was derived from all five included studies [18,19,20,21,22] and five descriptive themes. The peace and quietness of nature and the garden were described as having a positive, calming impact on the participants’ state of mind. The environment was experienced as calming and stress-reducing and helped the patients to slow down; to feel balanced, safe, and part of a whole; and also to become “one with nature”.

The participants expressed that the garden contributed to increased joy in daily tasks and that the sensory experiences of being in the garden helped them to be in the present and cope with daily challenges. A growing sense of coherence and belonging during the NBR was described, helping the participants to find meaning and values, and also that nature contributed to a state of mind that enabled new ideas and perspectives. Becoming one with nature through the NBR enabled them to get closer to their feelings, and the sensory experiences of the garden helped them to be in the present.

#### 3.5.2. Needs Being Met

The second analytical theme was derived from all five included studies [18,19,20,21,22] and five descriptive themes. The garden was described as giving a sense of safety and security and a feeling of being embraced. Participants spoke of how being in the garden met their needs, reflecting and harmonizing with their moods. Nature was perceived as representing their own needs. However, the garden also contributed to a sense of freedom, and was considered to be an undemanding, supportive, and permissive setting, allowing the participants to do nothing and to express their feelings without being judged. Nature was regarded as a restorative environment which enabled the participants to adjust to a slower pace in everyday life. The contact with nature during garden activities made them feel more vital, increased their well-being, and was described as consolidating the effects gained during other therapies. The slow rhythm of nature helped the participants to get a perspective on time and inspired them to redefine their life rhythm, which was perceived as beneficial for their mental health.

#### 3.5.3. Gaining New Insights

The third analytical theme was derived from all five included studies [18,19,20,21,22] and three descriptive themes. It describes insights gained by the participants in NBR. There were insights regarding themselves, such as increased self-awareness of destructive patterns and less constructive approaches to tasks in daily lives, and an improved understanding of their own needs. By creating a sense of kinship with nature, NBR also contributed to increased self-acceptance, increased acceptance of illness, and less self-judgement. This self-acceptance was perceived as improving the participants’ understanding of themselves, distancing them from performance-based values, and gaining better patience with themselves and their life circumstances. The participants also described insights of the garden and nature as being sources of inspiration. Participating in NBR stimulated creativity which enabled new ideas. It was as also experienced as a source of energy, which improved the participants’ inner strength. The garden was perceived as an accessible and useful arena with many possibilities.

#### 3.5.4. Personal Growth

The fourth analytical theme was derived from all five included studies [18,19,20,21,22] and three descriptive themes. It concerns positive changes that the participants achieved as a consequence of participating in NBR, contributing to personal growth. They considered themselves and their situation from new perspectives and gained a new view of life with openness to new opportunities. The simple act of planting, watering, and caring for plants was experienced as important for the participants’ self-esteem, and seeing lots of flowers created a sense of happiness. The participants also developed new constructive strategies and approaches to cope with needs and difficult situations in daily life. They described gaining self-efficacy and feeling empowered, which enabled them to move forward in life, break boundaries, and test new strengths.

## 4. Discussion

The main findings of this systematic review were that experiences and perceived effects of participating in NBR and spending time in a nature environment were described as positive for recovery, and that nature and the garden helped the patients slow down and feel calm, safe, and part of a whole. The environment met their needs and increased their self-awareness and self-acceptance, and was considered a source of energy and creativity. It promoted the development of new perspectives and approaches in daily life, which made them feel empowered to go forward.

The qualitative synthesis, aiming to describe experiences and perceived effects of the nature and garden components of NBR of the participants in the five qualitative studies, resulted in 16 descriptive themes organized under the four broad themes: Instilling calm and joy, Needs being met, Gaining new insights, Personal growth. Nine of the descriptive themes were based on explicit results from at least four of the five studies, strengthening our confidence that the findings are a reasonable representation of the phenomenon of interest. Even though the evidence base for the experienced benefits is small, the qualitative data were relevant, adequate, and highly consistent, supporting that the conclusions made are appropriate. For most findings, we had no or minor concerns regarding methodological limitations, relevance, coherence, and adequacy of the data.

Other reviews, of other types of outdoor, nature-based programs and/or in other populations, have reached similar conclusions regarding the positive experiences and perceived benefits of participating in a nature-based program [24,25,26]. A recent qualitative systematic review on nature-based therapeutic recreation showed that persons living with mental illness perceive these outdoor interventions as enjoyable and that therapeutic recreation makes a positive contribution to mental health [24]. Unlike our review, the review by Picton et al. [24] investigated therapeutic recreation programs, including sports and adventure activities, in various mental illness populations. They specifically identified therapeutic nature-based recreation as a socially inclusive and psychologically safe context that enhances the formation of social relationships and meaningful connections. The unique composition of NBR combines the impact of being in the garden and nature with established rehabilitation modalities such as physiotherapy, behavioral medicine, and occupational therapy.

Many of the descriptive themes that emerged from our synthesis may well be representative of the experiences of NBR by patients with other long-standing mental health problems, not explicitly stress-related. Even though some descriptive themes relate to strong experiences of stress relief in the garden and nature, other themes point at more general experiences of an environment that promotes mental health and well-being. Patients with depression and anxiety have described partly similar experiences, that participating in NBR contributed to a sense of being present and enabled personal growth [27]. Our findings also are in line with the findings of a qualitative study of patients with long-standing mental illness participating in NBR, showing that the garden was perceived as an undemanding setting that brings a sense of calm and safety and enables the participants to be in the present [28]. It is not unlikely that patients with other long-standing mental disorders would experience similar benefits, but this needs to be explored in future studies.

Although the overall evidence is scarce for statistically significant health effects in quantitative measures of NBR, the results of this qualitative synthesis indicate that in patients’ own views, positive health effects are experienced after participating in NBR. Our findings suggest that NBR could be a beneficial treatment option for patients with stress-related disorders, particularly in patients where other rehabilitation has been unsuccessful and where it is desirable to achieve one or more of the benefits that patients described in the included studies and that were perceived as specifically related to the garden/nature setting of the intervention. This implication is corroborated by Sahlin et al. [7], who found that NBR could enhance a stalled rehabilitation process in patients where initial care had been insufficient.

The inclusion criteria regarding the patients’ mental health conditions could be discussed since they are not in total agreement amongst the five studies. Our aim to look specifically at populations suffering from long-standing stress-related mental disorders, with the exclusion of PTSD, also involves some difficulties in defining the diagnoses to be included. For example, Exhaustion disorder is now often used in Sweden [21] for cases that in other countries probably would have been diagnosed with stress-related depression or anxiety. Comorbidity or overlap between these diagnostic entities is also common [29]. The Danish study [22] included patients on long-term sick-leave diagnosed with Reaction to severe stress or Adjustment disorder, and it is likely that many of these would have met the clinical criteria for Exhaustion disorder. Despite these differences in diagnosing stress-related mental disorder, we consider the patients in the five included studies to be sufficiently homogenous to draw conclusions of experiences of NBR in this systematic review.

A limitation with this review is that the systematic literature search only yielded five qualitative studies that met the inclusion criteria. They represented study populations from three NBR facilities in two Scandinavian countries, and thus the results are based on a rather limited set of qualitative data. It cannot be ruled out that studies in other countries and settings could have provided somewhat different results. 

The literature search was restricted to publications in English and Scandinavian languages, and thus studies published in other languages would not have been retrieved. Some methodological limitations were found in the included studies, mainly regarding the description of participants and researcher preunderstanding. One of the included studies used a recruitment method where the director of the NBR center selected the participants for the interviews [18], and another study performed 25 interviews but chose only to include 11 [21]. Both approaches could unwittingly have led to participants with positive experiences of NBR being overrepresented in the material. 

Furthermore, the number of participants was low (five) in the study by Adevi et al. [18]. However, no finding in this review was based on the study by Adevi et al. alone; the experiences from the five participants were supported by participants in at least two of the other included studies.

A strength of this review is that it follows a previously described and widely used method for qualitative synthesis [17] and the standardized CERqual approach for assessment of confidence in the evidence [12]. The quality of the included studies was assessed as moderate to high for four of the five studies. However, the confidence in the evidence for the review findings was moderate to low, mainly due to the low number of included studies. More qualitative studies of experiences of NBR in patients with stress-related mental disorders are warranted to increase the scientific evidence base. 

Broadening the inclusion criteria to include other disorders, such as depression and anxiety, and other nature-based interventions, might have yielded a larger sample and thus increased the confidence in the evidence for the findings. The population in this review was narrowed down to patients with stress-related disorders who have participated in NBR since we found it more clinically relevant to draw conclusions for a homogenous group of patients and type of treatment.

The similarities in terms of, e.g., population and NBR setting, in the five studies could also be regarded as a strength in the sense that they make it more likely that the results could be transferred to other patients with long-standing stress-related mental illness.

Future studies of patients with similar stress-related conditions taking part in NBR in other countries and contexts, as well as studies of patients with other common mental disorders, will show whether their experiences differ from those presented in this review.

## 5. Conclusions

This qualitative synthesis suggests that patients with long-standing stress-related mental disorders experience positive health effects after participating in NBR. However, confidence in the evidence for the findings was assessed as moderate to low. The findings suggest that NBR could be a beneficial treatment option for patients where other rehabilitation has been unsuccessful and where it is desirable to achieve one or more of the benefits that patients have experienced in these studies and that were specifically related to the garden/nature setting of the intervention. As the findings related to patients’ experiences are not congruent with findings from effectiveness studies, our review highlights the need for further research into objectively measured benefits of NBR as well as more studies of patient experiences of NBR among patients with stress-related mental disorders as well as in other populations.

## Figures and Tables

**Figure 1 ijerph-18-06897-f001:**
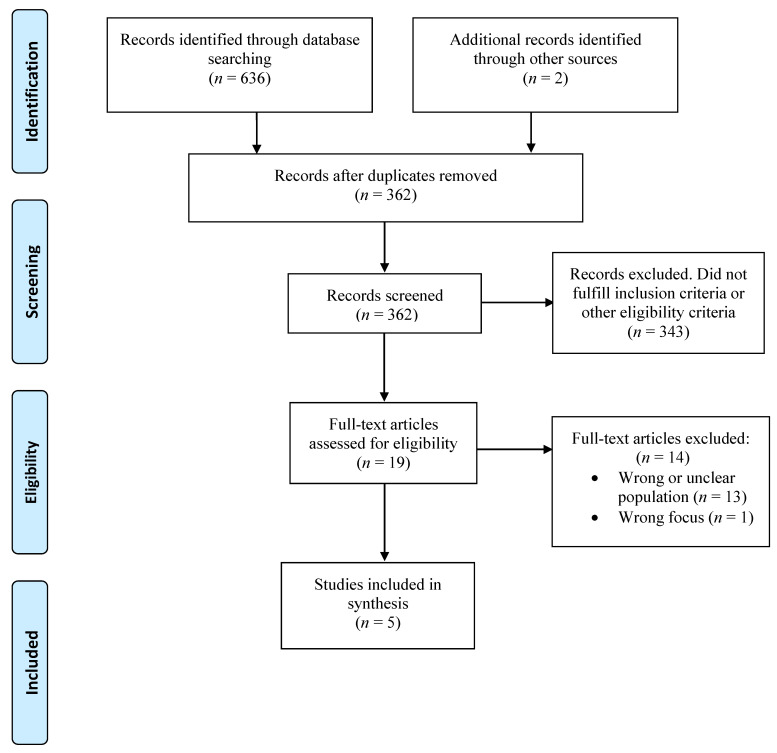
Flow diagram of the selection process (modified from Moher et al., 2009 [23].).

**Table 1 ijerph-18-06897-t001:** Characteristics of included studies.

Author, Year, Country	Aim	Setting	Length of NBR	Patients (*n*)	Mean Age (years)	Study Population
**Adevi,** **2013, Sweden** **[18]**	To explore the impact of garden therapy on stress rehabilitation, with special focus on the role of nature.	The health garden on the campus of the Swedish University of Agricultural Sciences in Alnarp, Sweden.	Not stated	5 (4 women, 1 man)	Range 25–60	Patients with exhaustion disorder
**Palsdottir,** **2014a,** **Sweden** **[19]**	To describe and assess changes in the participants’ experienced value of everyday occupations after NBR.	The health garden on the campus of the Swedish University of Agricultural Sciences in Alnarp, Sweden.	12 weeks	21 (19 women, 2 men)	47	Patients with adjustment disorder, reaction to severe stress, or depression
**Palsdottir,** **2014b,** **Sweden** **[20]**	To explore and illustrate how participants with stress-related mental disorders participating in NBR experience and describe their rehabilitation process.	The health garden on the campus of the Swedish University of Agricultural Sciences in Alnarp, Sweden.	12 weeks	43 (35 women, 8 men)	46	Patients on long-term sick leave for adjustment disorder, reaction to severe stress, or depression
**Sahlin,** **2012,** **Sweden** **[21]**	To explore how participants in a NBR program experience, explain, and evaluate their rehabilitation.	The Botanical Garden in Gothenburg, Sweden.	12–44 weeks	11 (8 women, 3 men)	43	Patients employed within administration and healthcare with Exhaustion disorder and/or depression and anxiety
**Sidenius,** **2017,** **Denmark** **[22]**	The aim of this study is to describe the phenomenon of participants’ lived experiences of the NBT in Nacadia during the course of a 10-week NBT program.	The therapy garden Nacadia, Copenhagen, Denmark.	10 weeks	14 (Women/men not reported)	Not reported	Patients with inability to work for at least 3 months with adjustment disorder and/or reaction to severe stress

NBR = Nature-based rehabilitation.

**Table 2 ijerph-18-06897-t002:** Assessment of methodological quality of the included studies.

Study	Methods of Data Collection	Methods of Data Analysis	Assessment of Methodological Quality (Based on the sbu Assessment Tool for Qualitative Studies)								
			Aim Well Defined	Sample Relevant, Selection and Context Well Described, Relevant Ethical Considerations	Researcher-Participant Relation Well Described	Data Collection Well Described and Relevant	Data Saturation	Researcher Pre-Understanding	Data Analysis Well Described	Findings Logical, Intelligible, Well Described	Findings Related to Theoretical Frame of Reference
Adevi 2013 [18]	Semi-structured interviews 0.5 to 1.5 years after the intervention	Grounded theory	Yes	Sample relevant but potentially biased; ethical considerations missing	No	Yes	Unclear	No	Yes	Yes	Yes
	Overall assessment: low-to-moderate					Limitations: Potential risk of bias in selection strategy (participants handpicked by garden manager), ethical considerations missing, preunderstanding not described					
Palsdottir 2014a [19]	Semi-structured interviews 12 weeks after the intervention	Qualitative content analysis	Yes	Yes	Unclear	Yes	Unclear	No	Yes	Yes	No
	Overall assessment: moderate					Limitations: No citations, interviews not recorded, preunderstanding not described/handled					
Palsdottir 2014b [20]	Semi-structured interviews within one month after program	Interpretative phenomenological analysis	Yes	Yes	Yes	Yes	N/A	Yes	Yes/unclear	Yes	
	Overall methodology assessment: moderate-to-high					Limitations: Appropriateness of analysis method unclear					
Sahlin 2012 [21]	Semi-structured interviews	Interpretative phenomenological analysis	Yes	Yes/ Ethical considerations missing	No	Yes	Yes	No	Yes	Yes/no	Yes
	Overall methodology assessment: moderate					Limitations: Ethical considerations missing, sample not clearly described, findings not clearly described					
Sidenius 2017 [22]	Semi-structured interviews, avg. 20 min, in 2nd, 5th, 9th week of program	Reflective life world research	Yes	Yes	No	Yes	Yes	Unclear	Yes	Yes	Yes
	Overall methodology assessment: high					Limitations: Lack of description of sample characteristics and preunderstanding in relation to data collection					

**Table 3 ijerph-18-06897-t003:** Summary of review findings and CERQual assessment.

Summary of Review Finding	Number of Studies Contributing to the Review Finding/Total Studies Included in the Review	CERQual Assessment of Confidence in the Evidence	Explanation of CERQual Assessment
**Instilling calm and joy**			
Patients described how nature’s peace and quiet had a calming impact on their state of mind.	5/5 [18,19,20,21,22]	Moderate confidence	Minor concerns about adequacy as the data come from a small number of studies. The studies were of moderate to high quality.
Patients expressed that NBR made them feel joy in daily tasks.	4/5 [18,19,20,22]	Moderate confidence	Minor concerns about adequacy as the data come from a small number of studies. The studies were of moderate to high quality.
Patients experienced that NBR created a sense of belonging to a greater whole and helped patients to find meaning and values.	4/5 [18,20,21,22]	Moderate confidence	Minor concerns about adequacy as the data come from a small number of studies. The studies were of moderate to high quality.
Patients described themselves as becoming one with nature during NBR, enabling them to get closer to their feelings.	3/5 [18,20,21]	Low confidence	Moderate concerns about adequacy as the data come from a very small number of studies and the studies were of moderate to high quality.
Patients described how sensory experiences in the garden helped them to be in the present.	3/5 [20,21,22]	Low confidence	Moderate concerns about adequacy as the data come from a very small number of studies. The studies were of moderate to high quality.
**Needs being met**			
Patients experienced that the garden gave a sense of safety and security.	4/5 [18,20,21,22]	Moderate confidence	Minor concerns about adequacy as the data come from a small number of studies. The studies were of moderate to high quality.
Patients experienced that the therapeutic garden met their needs.	3/5 [18,20,22]	Low confidence	Moderate concerns about adequacy as the data come from a very small number of studies, and the finding was seen in three of the five studies of which only one provided sufficiently rich data. The studies were of high quality.
The garden was perceived as an undemanding, tolerant, and permissive setting.	4/5 [18,20,21,22]	Moderate confidence	Minor concerns about adequacy as the data come from a small number of studies. The studies were of moderate to high quality.
Patients described that NBR helped them slow down and adjust to nature’s slower pace.	4/5 [18,19,20,21,22]	Moderate confidence	Minor concerns about adequacy as the data come from a small number of studies. The studies were of moderate to high quality.
Nature was experienced as a restorative environment that facilitated recovery.	5/5 [18,19,20,21,22]	Moderate confidence	Minor concerns about adequacy as the data come from a small number of studies. The studies were of moderate to high quality.
**Gaining new insights**			
Patients described gaining self-acceptance through kinship with nature, which helped them come to terms with being ill.	3/5 [18,21,22]	Low confidence	Moderate concerns about adequacy as the data come from a very small number of studies. The studies were of moderate to high quality.
Patients experienced that NBR increased their awareness of own needs and destructive patterns in daily life.	2/5 [19,22]	Low confidence	Moderate concerns about adequacy as the data come from a very small number of studies. The studies were of high quality.
Nature was perceived as a source of creativity and energy, making room for new ideas and affecting inner strength.	5/5 [18,19,20,21,22]	Moderate confidence	Minor concerns about adequacy as the data come from a small number of studies. The studies were of moderate to high quality.
**Personal growth**			
Patients’ experiences in nature and the garden helped them see things differently and develop new perspectives.	3/5 [18,21,22]	Low confidence	Moderate concerns about adequacy as the data come from a very small number of studies. The studies were of moderate to high quality.
The patients developed new approaches to tasks in their daily life.	3/5 [19,21,22]	Moderate confidence	Minor concerns about adequacy as the data come from a small number of studies. The studies were of moderate to high quality.
NBR was perceived as increasing empowerment, which enabled the patients to move forward.	4/5 [19,20,21,22]	Moderate confidence	Minor concerns about adequacy as the data come from a small number of studies. The studies were of moderate to high quality.

NBR = Nature-based rehabilitation.

**Table 4 ijerph-18-06897-t004:** Analytical and descriptive themes from the qualitative synthesis.

Analytical Themes	Descriptive Themes
Instilling calm and joy	Calming impact of nature
	Joy in daily tasks
	Finding meaning and sense of belonging
	Being one with nature
	Being in the present
Needs being met	Garden giving a sense of safety and security
	Garden meeting needs
	Garden as an undemanding and permissive setting
	Adjusting to nature’s slower pace
	Nature as a restorative environment
Gaining new insights	Gaining self-acceptance
	Increased self-awareness
	Insights of nature as source of creativity and energy
Personal growth	Developing new perspectives
	Developing new approaches
	Moving forward through empowerment

NBR = Nature based rehabilitation.

## Data Availability

Data available on request.

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
