# Peer review of "Nature-Based Rehabilitation for Patients with Long-Standing Stress-Related Mental Disorders: A Qualitative Evidence Synthesis of Patients’ Experiences"

_ijerph, 2021, doi:10.3390/ijerph18136897_

Round 1
Reviewer 1 Report
The introduction is quite clear: you state the origin of the problem both in terms of the individual seeking help (work stress) and the gap in research (qualitative studies of nature based rehabilitation). You gently lead readers to the importance of this study. One further suggestion would be looking to eco-psychology (Gregory Bateson, Bill Plotkin), who provide a basis for looking at a pervasive sense of disconnection as a cause of stress--but this sort of addition would not be necessary.
Section 2, on method and materials, shows a careful, caring, and conscientious attitude toward the data that was critical and comprehensive.
Section 3.1: The flowchart provides a useful initial transition from section 2.
Table 1: I find the gender/age information to be interesting and appreciate that it was provided. The formatting made this slightly hard to follow in terms of formatting: would advise against centering, or putting faint lines to divide the study.
Table 2: You do an excellent job of organizing your data and showing the rigor with which you analyzed the studies in a way simple enough to be clearly and quickly assessed by the reader. Phenomenal work.
Table 3: You do an excellent job of presenting the data in a systematic, organized fashion that clearly indicates the points of comparison and rationale for judgment.
Table 4: I would almost consider switching the descriptive/analytical settings between "New State of Mind" and "Personal Growth," but in a way this is only because the term "new" comes up. There's a lot of overlap between all of these categories, and the discernment of the researchers in coming up with the analytical categories for ease of use is notable. I still felt this after reading through 3.4.1-3.4.4. This would be more stylistically important than a problem with the content.
The discussion and your conclusions--especially concerning the need for further research of NBR's effectiveness--seem warranted by the data included and the methods employed.
Author Response
Thank you for your valuable comments and suggestions. Below are your comments followed by our replies in italics.
The introduction is quite clear: you state the origin of the problem both in terms of the individual seeking help (work stress) and the gap in research (qualitative studies of nature based rehabilitation). You gently lead readers to the importance of this study. One further suggestion would be looking to eco-psychology (Gregory Bateson, Bill Plotkin), who provide a basis for looking at a pervasive sense of disconnection as a cause of stress--but this sort of addition would not be necessary.
Reply: Thank you for this very interesting suggestion. We agree that eco-psychology could be a perspective of relevance for nature’s impact of the human mind. However, as we do not want to stray from our focus on NBR, we feel that this discussion is outside the scope of our review.
Section 2, on method and materials, shows a careful, caring, and conscientious attitude toward the data that was critical and comprehensive.
Section 3.1: The flowchart provides a useful initial transition from section 2.
Table 1: I find the gender/age information to be interesting and appreciate that it was provided. The formatting made this slightly hard to follow in terms of formatting: would advise against centering, or putting faint lines to divide the study.
Reply: Thank you, we have now changed the formatting of table 1 according to your suggestion.
Table 2: You do an excellent job of organizing your data and showing the rigor with which you analyzed the studies in a way simple enough to be clearly and quickly assessed by the reader. Phenomenal work.
Reply: Thank you!
Table 3: You do an excellent job of presenting the data in a systematic, organized fashion that clearly indicates the points of comparison and rationale for judgment.
Reply: Thank you!
Table 4: I would almost consider switching the descriptive/analytical settings between "New State of Mind" and "Personal Growth," but in a way this is only because the term "new" comes up. There's a lot of overlap between all of these categories, and the discernment of the researchers in coming up with the analytical categories for ease of use is notable. I still felt this after reading through 3.4.1-3.4.4. This would be more stylistically important than a problem with the content.
Reply: Thank you for this thoughtful comment! We agree that there is some overlap between the categories, yet they are distinct enough to warrant their categorisation into separate themes. The “personal growth” categories are more related to actions and things like coping strategies, empowerment, and new ways of dealing with different challenges. In contrast, the “new state of mind” categories pertain more to the mental calmness, relaxation and joy found through the gardening tasks and nature-based rehabilitation. We have changed the label of the first theme to make this clearer.
The discussion and your conclusions--especially concerning the need for further research of NBR's effectiveness--seem warranted by the data included and the methods employed.
Reviewer 2 Report
The submitted manuscript aptly synthesizes the results of five qualitative studies investigating the effects of nature-based rehabilitation (NBR) programs for those suffering from long-term stress-based mental health problems, identifying four main pathways through which being in nature helped the participants (as reported by them). This could be very informative for practitioners (emphasizing the factor nature has to offer which deems most relevant for the client in question, e.g. presenting nature as undemanding for a client that collapses under expectations). Also, it deems me a very fruitful ground for a quantitative study corroborating what the present paper distills from previous qualitative interviews, i.e. the benefits nature has to offer to people suffering from stress-induced mental health problems.
Seeing this great merit for both research and practitioner communities, I still have doubts about whether these authors should be allowed to publish a synthesis of only 5 studies mainly conducted by researchers other than the authors…. if I was the editor of this journal, I guess I would like to get consent of the (first) authors of the primary studies, as the present publication might decrease their citations, despite the main work of doing the intervention, recruiting and interviewing participants, and evaluating the data has been done by the original publications’ authors. I am not sure if the value added by the presented manuscript is so great in comparison of the original studies that it is acceptable to „claim“ their data for a new publication. I looked into the paper with the biggest sample, Palsdottir, Persson, Persson, & Grahn, 2014 - most of the insight discussed in the present manuscript is already in there, HOWEVER, much less concise and „usable“, thus, I am conflicted as to recommend publication with minor revisions or rejection.
I am not sure whether the CERQ criteria even can be allowed for such a low number of studies?
My main question probably revolves around the question whether the number of studies included in the manuscript could be incereased? I found it frustrating to go through so much detail in both abstract (actually, in lines 20-25, it seems a bit too much detail for an abstract) and method about the selection process of studies, just to end up with only 5 studies all stemming from largely the same geographic region, with substantial overlap between the authors of the mentioned studies. This raises even questions about the adequacy of the selection process - is it maybe a matter of terminology, „nature-based rehabilitation“, that is only used in Sweden and Denmark? Is it unthinkable to include more studies, e.g. „qualitative studies assessing the effect of nature (broader conceptualization of nature-based rehabilitation) on people that are stressed (broadening the target population to anyone struggling with stress) or people suffering from (any) mental health problem? Or could it even be combined with findings from quantitative studies, where e.g. NBR increased values on questionnaires x, y, or z, which in their items describe the topics of, e.g., restoration of energy or attention etc. What about the 19 studies that were excluded in the penultimate step? Is it unthinkable to include them in the final analyses?
I do not think this is an absolute necessity. The paper has merit in and of itself. However, I read it more as a lab-internal meta-analysis for purposes of designing an upcoming line of studies, a first chapter describing and explaining the foundation on which a corroboratory study is deriving its hypotheses from. I would very much hope the authors do this next step. I understand it need not be done, or could maybe not be done, in one paper. But I would very much like to see the next steps resulting from the information so neatly summarised in the manuscript’s present form.
If the paper were to be accepted, I would like to see Table 3 to more prominently feature the number of studies that showed the described finding, to get a quicker glimpse of how common it is, e.g., 5/5 after the naming of each finding in column 1, or if it has to be in the last column, then maybe in bold and numbers instead of text - generally, I applaud the use of the CERQ system, however, I could see the last column of table 3 becoming more readable if the sentences were a bit abbreviated or a coding scheme was used instead of full sentences.
Also, in Figure 1, PICO is referred to, without it being mentioned anywhere else in the manuscript.
All these points notwithstanding, I think the authors did a terrific job in synthesising information of qualitative research, which is valuable and difficult, so congratulations and thank you.
Author Response
Thank you for your valuable comments and suggestions. Below are your comments followed by our replies in italics.
The submitted manuscript aptly synthesizes the results of five qualitative studies investigating the effects of nature-based rehabilitation (NBR) programs for those suffering from long-term stress-based mental health problems, identifying four main pathways through which being in nature helped the participants (as reported by them). This could be very informative for practitioners (emphasizing the factor nature has to offer which deems most relevant for the client in question, e.g. presenting nature as undemanding for a client that collapses under expectations). Also, it deems me a very fruitful ground for a quantitative study corroborating what the present paper distills from previous qualitative interviews, i.e. the benefits nature has to offer to people suffering from stress-induced mental health problems. Seeing this great merit for both research and practitioner communities, I still have doubts about whether these authors should be allowed to publish a synthesis of only 5 studies mainly conducted by researchers other than the authors…. if I was the editor of this journal, I guess I would like to get consent of the (first) authors of the primary studies, as the present publication might decrease their citations, despite the main work of doing the intervention, recruiting and interviewing participants, and evaluating the data has been done by the original publications’ authors. I am not sure if the value added by the presented manuscript is so great in comparison of the original studies that it is acceptable to „claim“ their data for a new publication. I looked into the paper with the biggest sample, Palsdottir, Persson, Persson, & Grahn, 2014 - most of the insight discussed in the present manuscript is already in there, HOWEVER, much less concise and „usable“, thus, I am conflicted as to recommend publication with minor revisions or rejection.
Reply: Thank you for this reflection. It is not uncommon for reviews to include few studies, we have even seen review including 2 or even 0 studies (!). Even that is a result in itself, pointing to the paucity of research in the field. When you set out to do a review you can never know in advance how many studies that will meet your inclusion criteria and can be included. Furthermore, in qualitative research, it is never the quantity that matters (in original studies, e.g. the number of informants, in this case, the number of included studies), but rather the richness of the data.
We do not believe that for an original study to be included in a review would reduce citations – this is not the case in our own experience although we do not have data to either support or reject that hypothesis. On the contrary, we believe that for a study to be included in a review increases its exposure and may lead to more citations. The strength of a review/qualitative synthesis is that it synthesises findings from the original studies and goes beyond their individual results, thereby adding value and strengthening the evidence base on the topic (which in turn will add value to the original studies!).
I am not sure whether the CERQ criteria even can be allowed for such a low number of studies?
Reply: To the best of our knowledge, there are no lower limit for number of studies in a CERQual assessment. As can be seen in Table 3, the confidence in the evidence based on the CERQual assessment is made with regards to concerns about adequacy, due to the data coming from a small number of studies.
My main question probably revolves around the question whether the number of studies included in the manuscript could be incereased? I found it frustrating to go through so much detail in both abstract (actually, in lines 20-25, it seems a bit too much detail for an abstract) and method about the selection process of studies, just to end up with only 5 studies all stemming from largely the same geographic region, with substantial overlap between the authors of the mentioned studies. This raises even questions about the adequacy of the selection process - is it maybe a matter of terminology, „nature-based rehabilitation“, that is only used in Sweden and Denmark? Is it unthinkable to include more studies, e.g. „qualitative studies assessing the effect of nature (broader conceptualization of nature-based rehabilitation) on people that are stressed (broadening the target population to anyone struggling with stress) or people suffering from (any) mental health problem? Or could it even be combined with findings from quantitative studies, where e.g. NBR increased values on questionnaires x, y, or z, which in their items describe the topics of, e.g., restoration of energy or attention etc. What about the 19 studies that were excluded in the penultimate step? Is it unthinkable to include them in the final analyses?
Reply: Thank you for your comments and suggestions. We agree that the methods sections in both abstract and main text involves much detail, but this is the nature of systematic reviews/evidence syntheses, where transparency and detailed descriptions of what you have done is important. Nevertheless, we have shortened the methods section in the abstract. As stated above, the same rigorous process needs to be followed, regardless of the number of studies that finally are included in the review. We did not know from the outset how many studies we would be able to include, but (as noted above) believe that a review and synthesis greatly increases the value of the original studies and strengthens the evidence base.
From a clinical perspective we found it to be more relevant to narrow the population and intervention to be homogenous, rather than broadening it to include other disorders and similar interventions. We have added a few sentences about this in the Discussion line 355.
I do not think this is an absolute necessity. The paper has merit in and of itself. However, I read it more as a lab-internal meta-analysis for purposes of designing an upcoming line of studies, a first chapter describing and explaining the foundation on which a corroboratory study is deriving its hypotheses from. I would very much hope the authors do this next step. I understand it need not be done, or could maybe not be done, in one paper. But I would very much like to see the next steps resulting from the information so neatly summarised in the manuscript’s present form.
Reply: Thank you for your comment. We agree, the results of this review warrant further studies within the area of NBR and stress-related mental disorders and/or nearby conditions, and it could certainly provide the basis for future studies. We mention implications for future research in the discussion and conclusion sections.
If the paper were to be accepted, I would like to see Table 3 to more prominently feature the number of studies that showed the described finding, to get a quicker glimpse of how common it is, e.g., 5/5 after the naming of each finding in column 1, or if it has to be in the last column, then maybe in bold and numbers instead of text - generally, I applaud the use of the CERQ system, however, I could see the last column of table 3 becoming more readable if the sentences were a bit abbreviated or a coding scheme was used instead of full sentences.
Reply: Thank you for your suggestion, we have now tried to make this information clearer in Table 3, column 2. We have also shortened the text in column 4.
Also, in Figure 1, PICO is referred to, without it being mentioned anywhere else in the manuscript. Thank you for having noticed this – we initially structured our inclusion criteria according to the PICO format, but changed it to the PEO format, described in the Methods section. We have corrected the figure.
All these points notwithstanding, I think the authors did a terrific job in synthesising information of qualitative research, which is valuable and difficult, so congratulations and thank you.
Reply: Thank you!
Reviewer 3 Report
This paper has potential to make practical contributions. Below, I would like to suggest a few points that might be helpful for future revision. 1. Line 89: I am curious if qualitative studies are the only ways to assess benefits of NBR interventions. Is it at all possible to evaluate the effect of NBR with quantitative measures? Why is it that there are limited studies that objectively measure NBR’s quantitative effect (line 85)? It would be more helpful if the authors could provide some explanation about the value of qualitative studies, rather than quantitative ones, in proving the effect of NBR. 2. Line 100: What does ‘HTA’ stand for? Also, what do ‘ENTREQ’ (line 102) and ‘PTSD’(line 114) stand for? 3. Line 127: How many reviewers were there in total? 4. Line 152: Please explain in detail the coding scheme used in the analysis. More specifically, what was the scheme for the ‘line-by-line coding’ (line 197) that resulted in 178 preliminary codes? 5. Line 278: If the benefits of NBR are similar to other types of outdoor, nature-based programs, what might be the benefits specific to NBR that other types of programs cannot offer? The study contribution would be much clearer if the authors could explain specific benefits of NBR that are distinct from other kinds of nature-based programs. 6. Line 354: As the authors noted, if confidence in the evidence for the findings was assessed as moderate to low, would it be fine to rely on these studies as main evidence of the current study?Author Response
Thank you for your valuable comments and suggestions. Below are your comments followed by our replies in italics.
This paper has potential to make practical contributions. Below, I would like to suggest a few points that might be helpful for future revision.
- Line 89: I am curious if qualitative studies are the only ways to assess benefits of NBR interventions. Is it at all possible to evaluate the effect of NBR with quantitative measures? Why is it that there are limited studies that objectively measure NBR’s quantitative effect (line 85)? It would be more helpful if the authors could provide some explanation about the value of qualitative studies, rather than quantitative ones, in proving the effect of NBR.
Reply: Thank you for this suggestion. However, we feel that we already have covered this aspect on line 86 -93. - Line 100: What does ‘HTA’ stand for? Also, what do ‘ENTREQ’ (line 102) and ‘PTSD’(line 114) stand for?
Reply: HTA is explained on line 81. ENTREQ is explained on line 103, just before the abbreviation. We have added the explanation for PTSD (Post-traumatic stress disorder) at its first mention in the manuscript, line 115. - Line 127: How many reviewers were there in total?
Reply: Thank you for this question which made us realize that this information was missing in the text. We have added this information in the manuscript at line 126 and 129. - Line 152: Please explain in detail the coding scheme used in the analysis. More specifically, what was the scheme for the ‘line-by-line coding’ (line 197) that resulted in 178 preliminary codes?
Reply: The coding scheme involved identifying text segments that were relevant to the study aim, and condensing these segments, line by line, into codes. We have described this in more detail in the Methods section (line 155 -160). - Line 278: If the benefits of NBR are similar to other types of outdoor, nature-based programs, what might be the benefits specific to NBR that other types of programs cannot offer? The study contribution would be much clearer if the authors could explain specific benefits of NBR that are distinct from other kinds of nature-based programs.
Reply: Thank you for this comment. We have now added a sentence about this in the discussion, line 296. - Line 354: As the authors noted, if confidence in the evidence for the findings was assessed as moderate to low, would it be fine to rely on these studies as main evidence of the current study?
Reply: Thank you for this thoughtful comment. The quality of the included studies was assessed as moderate to high for four if the five studies. However, the confidence in the evidence in the findings was moderate to low in this review mainly due to the low number of included studies. More studies are warranted to increase the scientific basis and the confidence in the evidence. We have now added three sentences about this in the discussion, line 355 – 365.
Round 2
Reviewer 2 Report
I can see the author did the minimal adjustment absolutely necessary.... as the other reviewers were okay with this, I will go along and suggest publication. However, I would really like to encourage the authors to just see this as a groundwork for future studies specifically testing the mechanisms behind NBR-benefits that are outlined so well in the present paper. Kudos for that.